# Shifting patterns of dengue three years after Zika virus emergence in Brazil

Francesco Pinotti [1] ✉, Marta Giovanetti [2,3,4], Maricelia Maia de Lima [5], Erenilde Marques de Cerqueira[5], Luiz C. J. Alcantara[2,3], Sunetra Gupta [1], Mario Recker [6,7] & José Lourenço[8]

In 2015, the Zika virus (ZIKV) emerged in Brazil, leading to widespread outbreaks in Latin America. Following this, many countries in these regions reported a significant drop in the circulation of dengue virus (DENV), which resurged in 2018-2019. We examine age-specific incidence data to investigate changes in DENV epidemiology before and after the emergence of ZIKV. We observe that incidence of DENV was concentrated in younger individuals during resurgence compared to 2013-2015. This trend was more pronounced in Brazilian states that had experienced larger ZIKV outbreaks. Using a mathematical model, we show that ZIKV-induced cross-protection alone, often invoked to explain DENV decline across Latin America, cannot explain the observed age-shift without also assuming some form of disease enhancement. Our results suggest that a sudden accumulation of population-level immunity to ZIKV could suppress DENV and reduce the mean age of DENV incidence via both protective and disease-enhancing interactions.

Dengue virus serotypes 1 to 4 (DENV1-4) are mosquito-borne flaviviruses of considerable public health concern. DENV has been estimated to cause almost 400 million infections globally every year[1]. In endemic and hyper-endemic areas, recurrent DENV epidemics are able to overwhelm hospitals, especially in resource-limited settings. Human infections may display a wide range of clinical manifestations, with the most severe forms leading to vascular leakage and shock[2]. Importantly, pre-existing cross-reactive antibodies may exacerbate disease following heterologous infections through antibody-dependent enhancement (ADE)[3,4]. It is also well-established that disease severity can vary significantly depending on the identity and sequence of infecting DENV serotypes[5–8].

Zika virus (ZIKV) is an emergent flavivirus with genetic proximity to DENV, with which it also shares some transmitting vectors, most notably *Aedes* mosquitoes[9,10]. First identified in 1947 in Africa, ZIKV emerged in Yap State in 2007, in French Polynesia in 2013–2014 and in the Americas (first detected in the Northeastern region of Brazil) in 2015[11]. Although the ensuing epidemics did not lead to any reported deaths, severe clinical outcomes, including Guillain-Barré syndrome and neonatal microcephaly, were associated with ZIKV infection[12–14]. After its introduction in 2013–2014[15] and the following two large epidemic waves in 2015–2016, ZIKV incidence in Brazil has dropped considerably, likely due to widespread accumulation of population immunity[16,17].

In Brazil, the 2015–2016 ZIKV epidemic was followed by two years of low DENV circulation, a phenomenon also observed across other South American countries and the Caribbean[18,19]. The resurgent outbreaks in Brazil were associated mostly with DENV1, which dominated everywhere in the country in 2015–2016, and DENV2, which replaced the former in the Midwest and the Southeast[19]. Several, non-mutually exclusive hypotheses have been proposed to explain this phenomenon, including changes in arbovirus surveillance and control and of

[1]Department of Biology, University of Oxford, Oxford, United Kingdom. [2]Laboratório de Flavivírus, Instituto Oswaldo Cruz, Fundação Oswaldo Cruz, Rio de Janeiro, Brazil. [3]Instituto Rene Rachou, Fundação Oswaldo Cruz, Belo Horizonte, Minas Gerais, Brazil. [4]Sciences and Technologies for Sustainable Development and One Health, University of Campus Bio-Medico di Roma, Rome, Italy. [5]Universidade Estadual de Feira de Santana, Feira de Santana, Bahia, Brazil. [6]Centre for Ecology and Conservation, University of Exeter, Penryn, United Kingdom. [7]Institute for Tropical Medicine, University of Tübingen, Tübingen, Germany. [8]Católica Biomedical Research, Católica Medical School, Universidade Católica Portuguesa, Lisbon, Portugal. ✉e-mail: francesco.pinotti@biology.ox.ac.uk

mosquito abundance[20]. One hypothesis of interest here is that transient cross-protection induced by the large ZIKV epidemic waves temporarily suppressed DENV before it resurged across most of the country in late 2018[21]. Due to the complex nature of DENV dynamics and the spatial variation in epidemic control and mosquito abundance, it has been difficult to reach a consensus on the drivers of DENV absence post-ZIKV.

Immune mechanisms that modulate susceptibility to infection and disease severity are well described in the literature and recognized to have important consequences on DENV epidemiology[8,22,23]. Given the genetic and antigenic similarities between ZIKV and DENV1-4, it is plausible that ZIKV can modulate the disease severity of dengue, as observed in sequential infections by DENV1-4 serotypes. Supporting this are the reports that the viruses are able to induce cross-reactive antibody responses to shared epitopes[24–26]. Anti-ZIKV antibodies have been shown to increase viraemia and disease severity in mice and macaques upon challenge with DENV[27–30], although other studies in macaques failed to suggest an effect[31–33]. Two hospital cohort studies in Nicaragua and Sri Lanka reported an increased risk of severe forms of dengue in ZIKV+ human patients[34,35]. Interestingly, both studies found that anti-ZIKV antibodies may enhance disease in both primary and secondary DENV infections, suggesting that ZIKV may not simply behave as another DENV serotype. In the latter case in fact, previous infection with DENV and ZIKV would be expected to stimulate a broadly neutralising anti-DENV response and thus protect against severe disease in later DENV infections. One possible explanation for this is that sequential exposure to DENV and ZIKV results in a limited amount of anti-DENV antibodies compared to heterologous DENV infections[34,36]. Alternatively, differences in antibody quality among cross-reactive and DENV-specific immune responses may also explain enhancement. A study of antibody kinetics found that secondary flavivirus infections yielded similar antibody titers irrespective of the infecting pathogen, thus underscoring the importance of antibody quality[36]. Alternatively, failure to stimulate a robust cellular immune response could also explain disease enhancement[37,38].

In this work, we aim to investigate the possible impact of ZIKV-induced cross-reactivity on DENV epidemiology. By analysing incidence and hospitalisation data from Brazil (2000–2019), we show that the post-ZIKV dengue resurgent epidemic was characterised by a significant reduction in the average age of infection compared to the pre-ZIKV period. Furthermore, we demonstrate that Brazilian states with higher ZIKV attack rates were associated with the largest shift in age of resurgent DENV infections. Using a mathematical transmission model, we assess the ability of distinct mechanisms of DENV-ZIKV immune interactions to explain the observed epidemiological data patterns. We show that while ZIKV-induced short-term protection can explain the temporary suppression of DENV, assuming some degree of disease enhancement appears necessary to capture the observed coinciding negative relationship between DENV age distribution and ZIKV attack rate during the resurgence of DENV after 2018.

## Results
### Larger shifts in age distribution of DENV are consistent with high circulation of ZIKV in Northeastern Brazil
The 2015–2016 ZIKV epidemic in Brazil was followed in many regions by a trough in DENV incidence. This is exemplified in Fig. 1 for the city of Feira de Santana (FdS), in the Northeastern state of Bahia. Notably, the trough was succeeded by a large dengue outbreak between late 2018 and early 2019, similar to that seen in other regions[18,19].

A closer inspection of DENV incidence data suggests that cases reported in the period 2018–2019 were typically younger compared to the pre-ZIKV era, which is shown in Fig. 2 for the cities of FdS and Salvador and the state of Bahia (where the previous cities are located).

We can quantify this shift in age by calculating the relative difference in the average age of DENV cases or hospitalisations before

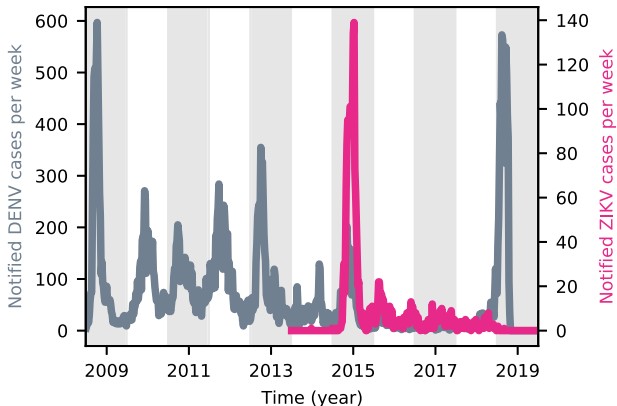

**Fig. 1 | DENV and ZIKV incidence in Feira de Santana.** Weekly notified DENV (gray) and ZIKV (fuchsia) cases in Feira de Santana, Bahia, from 2009 to 2019.

and after the emergence of ZIKV, here referred to as $\Delta_{cases}$ and $\Delta_{hosp}$, respectively (see Methods). As shown in Fig. 3A, B, states in the Northeastern region of Brazil experienced a shift towards younger ages ($\Delta < 0$), both for reported cases and hospitalisations (full distributions in Northeastern states are shown in Supplementary Fig. 1). According to recent modelling work, these states in the Northeast were particularly hit by ZIKV with attack rates above 40% in 8/9 states and above 60% in 4/9 states. In contrast, the vast majority of other states had lower attack rates, especially in the Southern region, and showed little to no difference in the average age of infection (Fig. 3C)[39].

To characterise the relation between the age-shift in DENV cases, $\Delta$, and ZIKV attack rates at the level of individual states, we fitted a linear model including ZIKV attack rate and other eco-epidemiological variables as covariates (Supplementary Fig. 2). Model inference revealed a strong, statistically significant association between ZIKV and $\Delta$, for both hospitalisations (regression coefficient: -0.44, 95% CI [-0.75, -0.13]) and cases (regression coefficient: -0.3, 95% CI [-0.46, -0.13]). The linear model did not support any significant association with the remaining covariates (Supplementary Fig. 2), including the relative proportion of DENV1 serotype, which dominated in the North and Northeast regions in 2019 (Fig. 3D), but was outcompeted by DENV2 elsewhere. Additionally, we find that the negative correlation between estimated ZIKV attack rate and $\Delta$ is robust with respect to the pre-ZIKV reference period used to compute the mean age of DENV $\bar{A}_{pre}$ (Supplementary Fig. 3), and that it can not be explained by historical trends in mean age of DENV (This analysis can be found in Supplementary Fig. 4, 5, 6). These findings further corroborate the existence of a change point in DENV epidemiology in Brazil associated with the spread of ZIKV.

### ZIKV-induced cross-reactivity can modify the age distribution of DENV cases
We next used a mathematical transmission model (see Methods) to investigate the potential implications of ZIKV-induced cross-reactions on post-ZIKV DENV epidemiology. The model describes the co-circulation of 4 DENV serotypes and the emergence of ZIKV in a host population that is calibrated to the Bahian city of Salvador in the Northeast (see Methods and Supplementary Fig. 9-12). We first considered the case where ZIKV infection temporarily prevents DENV transmission ($\gamma_{ZD} = 0$), but does not affect DENV disease severity. Model simulations revealed that if ZIKV-induced cross-protection is short-lived ($l_Z$ less than 1 year) DENV incidence in the post-ZIKV era experiences a temporary dip, followed by a few years of high transmission. While after ZIKV introduction waves DENV may circulate at low levels from 2 to 4 years, depending on the duration of cross-protection ($l_Z$), the initial ZIKV attack rate ($\rho_Z$) drives the magnitude of the perturbation (Supplementary Fig. 13), with higher levels of ZIKV

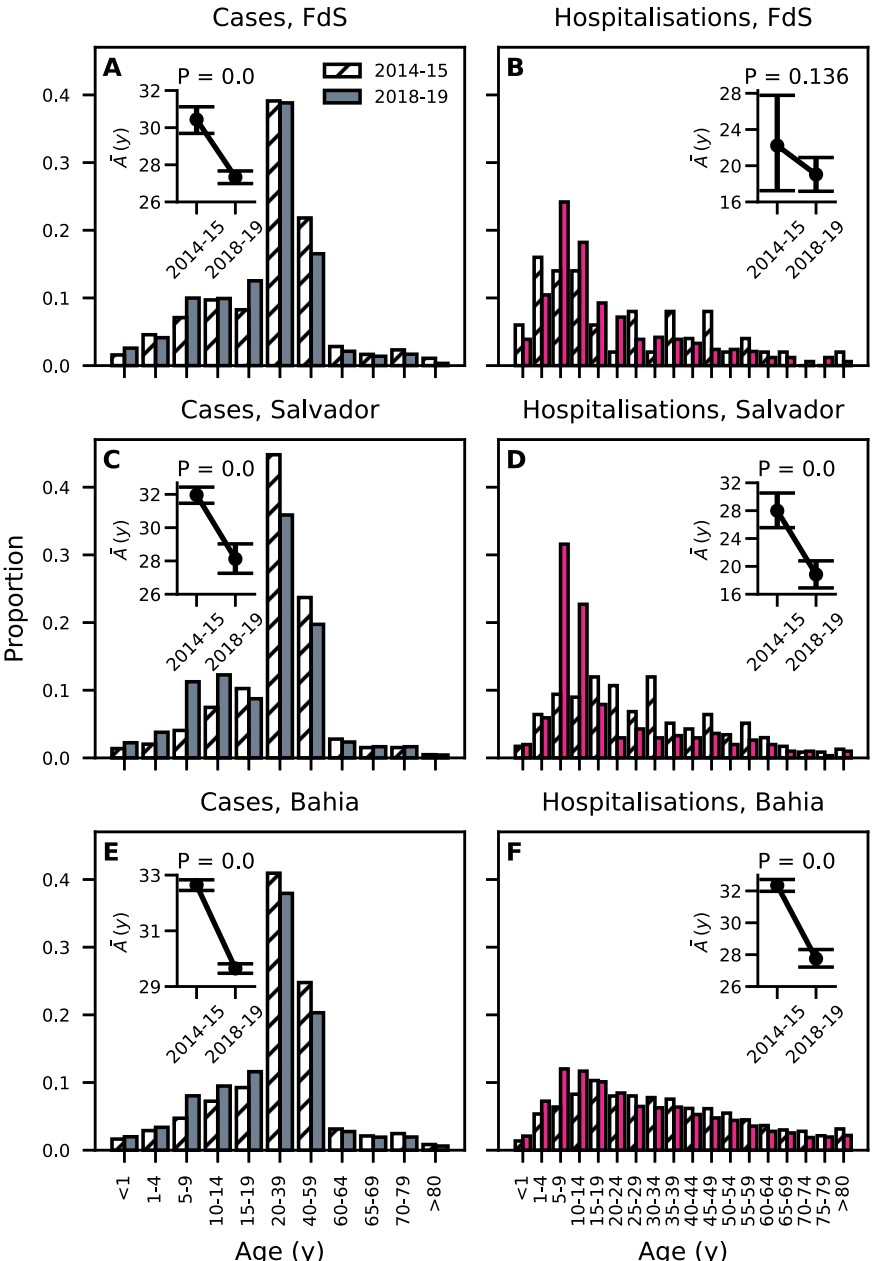

**Fig. 2 | Pre- and post-ZIKV DENV incidence age distributions in Feira de Santana, Salvador and Bahia.** Bars represent age distributions of DENV incidence before (years 2014–2015, hatched) and after (2018–2019, solid) ZIKV. Left and right columns display reported cases and hospitalisations, respectively. From top to bottom: Feira de Santana (**A, B**), Salvador (**C, D**) and Bahia (**E, F**). Insets compare median age of DENV incidence in 2014–15 and 2018–19. Error bars denote the 95% C.I. uncertainty around median estimates and were calculated using a bootstrap procedure (Methods). One-sided permutation tests with *n* = 100000 replicates (Methods) were performed to assess whether the mean age of DENV cases was significantly lower in the post-ZIKV period with respect to the pre-ZIKV period. Making an exception for panel **B**, incident cases reported in 2018–2019 were significantly younger compared to those reported in 2014-2015, *p* = 0). In **B**, the difference in mean ages was negative (−3.02 years) but not statistically significant (*p* = 0.136). Note that case and hospitalisation data make use of different age groupings.

seroprevalence rates causing lower DENV incidence. However, while in this case the model predicts a strictly negative impact of ZIKV attack rate on subsequent DENV transmission, available data appears to suggest a non-monotonic relationship (Supplementary Fig. 14).

In this scenario in which ZIKV does not affect disease severity in subsequent DENV infections, with respect to the age distribution of such DENV infections, simulated cases sampled during the resurgent DENV wave were found to be slightly older, not younger, in contrast with observed data (Fig. 4A, grey versus black lines). Furthermore, the age-shift in DENV cases (Δ) between pre- and post-ZIKV periods increases with increasing ZIKV attack rates, again contrasting observed

data. Mechanistically this happens because in the absence of any cross-regulation of DENV severity, the only effect of ZIKV is to delay individuals to experience their first or second DENV infections by a few years (thus increasing the mean age of DENV infection). Therefore, while short-term ZIKV-induced cross-protection is sufficient to explain the temporary disappearance of DENV, alone it falls short of explaining the coinciding reduction in the mean age of DENV cases in the post-ZIKV era.

We next explored the possibility that a previous infection with ZIKV modulates DENV disease severity in addition to offering short-term protection. For this, we tracked individual DENV cases in the

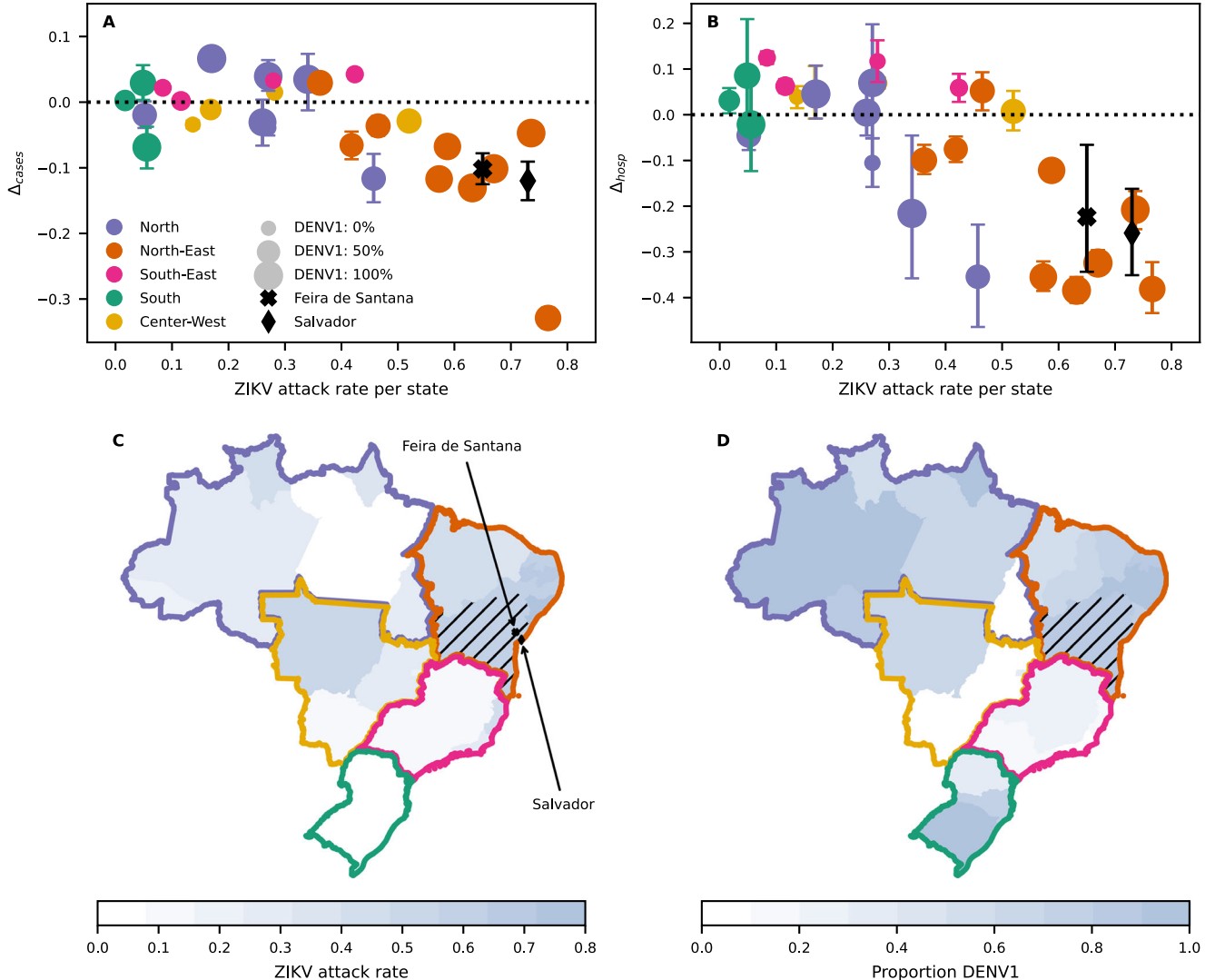

**Fig. 3 | Impact of ZIKV on DENV case age across Brazil. A, B** Relative age difference $\Delta$ between pre- and post-ZIKV DENV cases across individual states (dots) vs estimated ZIKV attack rate. Panels **A, B** refer to notified DENV cases and hospitalizations, respectively. Pre- and post-ZIKV mean case ages are computed by pooling age-specific incidence counts from the periods 2013-2015 and 2018-2019, respectively. Annual time series of age-shifts and mean age of incident cases are shown in Supplementary Fig. 7 and 8, respectively. Dot size is proportional to the fraction of DENV1 isolates collected between 2018 and 2019. Values corresponding to municipalities of Feira de Santana and Salvador are denoted with black markers; ZIKV seroprevalence for these two locations was taken from[17,59]. Error bars represent the 95% C.I. on $\Delta$. Mean and C.I. are estimated using 1000 bootstrap samples (Methods). **C** ZIKV attack rate per state. **D** Proportion of DENV1 serotype isolates per state in 2019. Black hatches highlight the state of Bahia, where Salvador and Feira de Santana are located. Maps in **C, D** were realised using GeoPandas[63] v0.11.1 using publicly available shapefiles obtained from the Brazilian Institute of Geography and Statistics[64].

model and weighted them according to their DENV and ZIKV serostatus to reflect possible, differential odds of severity, and hence likelihood of reporting (given a passive surveillance system). We investigated two potential mechanisms by which ZIKV-induced cross-reactivity may enhance disease in later DENV infections: (1) ZIKV effectively behaves as a fifth DENV serotype, enhancing disease in DENV-naive hosts, such that primary DENV infections in ZIKV+ are as severe as DENV secondary infections in ZIKV- hosts, and (2) ZIKV enhances disease in both primary and secondary DENV infections based on previous results from Katzelnick and colleagues[34]. In line with qualitative patterns observed across Brazilian states (Fig. 3), both assumptions lead to a shift ($\Delta$) towards younger age in post-ZIKV reported DENV cases with increasing ZIKV attack rates (Fig. 4A, orange and magenta). However, mechanism (1) displays the largest reduction in age (Fig. 4A, magenta) since it relies mainly on ZIKV+ hosts that are also DENV-naive and thus necessarily younger, whereas mechanism (2) displays a smaller reduction in age (Fig. 4A, orange) since it also

enhances disease in older cases (see Supplementary Fig. 15 for full examples of resulting age distributions).

In general, allowing for ZIKV+ individuals to experience varying levels of cross-protection to DENV infection depending on experiencing primary or secondary DENV infections ($w_0^+$ and $w_1^+$) can yield a range of outcomes for the shift in the age of DENV cases ($\Delta$) in the post-ZIKV era (Fig. 4B). In particular, we find that the differential contributions of primary and secondary DENV infections in ZIKV-positive individuals tend to affect the shift in age $\Delta$ in opposite directions: ZIKV-induced disease enhancement in primary DENV infections decreases mean DENV case age, while ZIKV-induced disease enhancement in secondary DENV infections leads to increases in mean DENV case age. The scenario most compatible with the observed reduction in DENV case age post-ZIKV (Fig. 3) is when enhancement of primary DENV infection ($w_0^+$) is sufficiently larger than enhancement of secondary DENV infection ($w_1^+$). This suggests a leading role of primary infections in the observed patterns, rather than secondary infections.

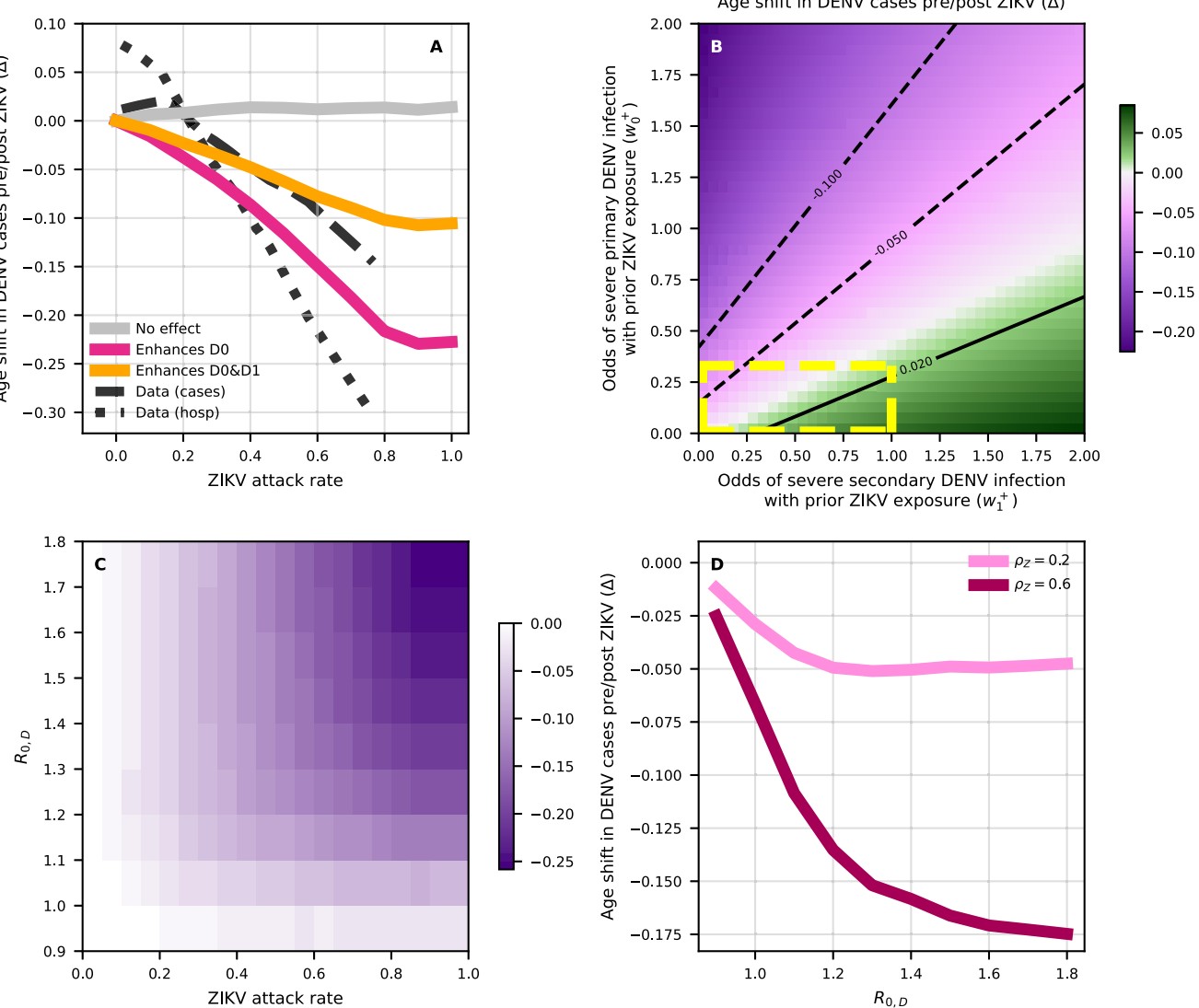

**Fig. 4 | Impact of ZIKV on age of DENV incident cases in a Brazilian community.** **A** Relative case age-shift ($\Delta$) as a function of ZIKV attack rate. $\Delta$ is calculated with respect to a scenario with no ZIKV and is based on reported incidence. The latter is obtained by weighting raw incidence with weights $w_n^{+/-}$ depending on the number $n = 0,1,2,3$ of previous DENV infections and seropositivity/negativity to ZIKV (+/-). Without loss of generality, we set $w_1^- = 1$ as a reference value. Other coefficients are $w_0^- = 0.33$, $w_2^- = 0.26$ and $w_3^- = 0$ based on findings from Katzelnick et al.[34]. (Light grey) ZIKV does not affect severity ($w_n^+ = w_n^-$ for any $n$). (Fuchsia) ZIKV enhances disease in primary DENV infections ($w_0^+ = w_1^- = 1$) but not in subsequent DENV infections ($w_1^+ = w_2^- = 0.26$, $w_{>1}^+ = w_{>2}^- = 0$). (Orange) ZIKV enhances disease in primary and secondary DENV infections ($w_0^+ = w_1^+ = 1$), but not in subsequent ones ($w_{>1}^+ = w_{>2}^- = 0$). Black lines correspond to case (dashes) and hospitalisation (dots) data. These were obtained by smoothing fitted values from Supplementary Fig. 2

with a lowess smoother. **B** Relative age-shift in new DENV cases as a function of odds of primary (y-axis) and secondary (x-axis) DENV cases being reported in ZIKV-positive hosts, while $w_2^+ = 0.26$, $w_{>2}^+ = 0$. Configurations where ZIKV is protective against both severe primary and secondary DENV infections (i.e. $w_0^+ < w_0^-$ and $w_1^+ < w_1^-$) are delimited by yellow dashes. ZIKV attack rate is set to 70%. In **A,B**, model parameters are calibrated as in Supplementary Fig. 9 and results are averaged over 100 simulations. **C** Relative age-shift as a function of ZIKV attack rate and $R_{0,D}$, assuming that ZIKV behaves as a fifth DENV serotype (see fuchsia line in **A**). **D** Lines show the average age-shift vs $R_{0,D}$ for selected values of ZIKV attack rate. In **C,D**, we let DENV spread for 100 years before introducing ZIKV. $\Delta$ is calculated using incidence from the fourth year since ZIKV introduction. Results are averaged over 300 simulations.

Immunomodulation in primary DENV infections would also explain the large contribution of paediatric infections (children aged 1-9 years old) to hospitalisations as shown in Fig. 2. Indeed, a further analysis suggests that a significant fraction of incident cases reported in 2018 and 2019 in this age group in the Northeast are expected to be primary DENV infections that had been exposed to ZIKV in 2015-2016 (Supplementary Fig. 16).

Finally, we assess the potential effect of background DENV population-level immunity during the pre-ZIKV era, on the observed shift in the age of DENV cases ($\Delta$) in the post-ZIKV era. To investigate this we considered a range of epidemiological scenarios, characterised by different values of DENV basic reproduction number ($R_{0,D}$) (Fig. 4C).

We find that the age-shift in DENV cases post-ZIKV depends significantly on the baseline level of DENV transmission. In particular, we observe almost no effect ($\Delta \approx 0$) when pre-ZIKV DENV circulation is low ($R_{0,D} < 1$), even in the extreme scenario where ZIKV infects the entire population ($\rho_Z = 1$). Increasing $R_{0,D}$ typically magnifies $\Delta$, regardless of whether DENV enhances disease in DENV-naive hosts only (mechanism (1)) or in both naive hosts and hosts with a single prior DENV infection (mechanism (2), Supplementary Fig. 17). In addition, the age-shift $\Delta$ decreases more steeply at low $R_{0,D}$ (for complete examples see Supplementary Fig. 18). Interestingly, the shifted age distributions always peak in the 5-14 years old under mechanism (1) for a range of $R_{0,D}$ values (Supplementary Fig. 18). However, older age groups also

become more pronounced in low-transmission settings, where the baseline mean age of DENV cases is higher. Likewise, the age-shift Δ is more sensitive to $R_{0,D}$ when the attack rate of ZIKV is higher (see Fig. 4D), suggesting a connection with the underlining transmission potential of both arboviruses, notably transmitted by the same mosquito species. Accordingly, we also find that a hypothetical positive association between ZIKV and DENV transmission due to, e.g., shared ecological and epidemiological factors, can further exacerbate the effect of ZIKV herd-immunity ($\rho_Z$) on post-ZIKV age-shifts of DENV cases (Δ) (Supplementary Fig. 19).

## Discussion

After two years of low circulation across Brazil and the Caribbean, DENV resurged in 2018-2019 in many Brazilian regions in a synchronous fashion. Notably, both serotypes 1 and 2 dominated this resurgence period while presenting discordant spatial distributions. In this work, we analysed the trends of reported dengue cases and hospitalisations across Brazilian states before and after the large, countrywide ZIKV outbreak of 2015-2016. Existing data showed that for most states, the typical age of reported DENV cases was significantly lower in the resurgent post-ZIKV period of 2018-2019 compared to the pre-ZIKV era. In parallel, data also suggested that this post-ZIKV reduction in DENV age was negatively associated with local ZIKV attack rates during 2015-2016.

Here, we modelled a range of immunological assumptions about ZIKV-DENV cross-reactivity and assessed their possible contribution to observed changes in dengue age distributions between the pre- and post-ZIKV eras. Our results show that a certain amount of disease enhancement during primary and/or secondary DENV infections can explain the observed shift towards younger ages of reported cases across Brazil in the post-ZIKV era. In contrast, assuming no enhancement or long-lived protection yielded age patterns that were at odds with observed data. To start understanding this finding, it is necessary to note that the large-scale ZIKV epidemic of 2015-2016 is expected to have affected age classes uniformly. Upon resurgence of DENV in 2018-2019, ZIKV+ hosts with zero or one prior DENV infection were necessarily younger than those that have experienced more DENV infections. The observed shifts to lower ages of DENV cases in the post-ZIKV era would then be compatible with ZIKV-induced enhancement of DENV happening precisely in those subgroups of the population with zero or one prior DENV infection. Future research work is required, however, to verify whether alternative mechanisms, e.g. enhancement of infection rather than disease, provide a better explanation of empirical observations. A corollary of our rationale is that larger ZIKV attack rates in 2015-2016 should mirror larger age-shifts of DENV cases post-ZIKV, which we found could explain observed state-to-state variations in terms of the observed age-shifts. Indeed, the Northeastern states (including Bahia) were at the epicentre of the 2015-2016 ZIKV epidemic, having experienced some of the largest ZIKV attack rates and the largest reductions in age of DENV infections in the resurgence of 2018-2019. It is possible, however, that current estimates of ZIKV attack rates underestimate true ZIKV circulation due to misdiagnosis between Zika and dengue[40]. Despite these concerns, our results should still hold provided that the extent of underestimation is not extremely heterogeneous across Brazil.

Although our results imply a role of ZIKV in exacerbating dengue pathogenesis, the data and our model suggest that this follows a period of cross-protection. The near absence of dengue in the wake of the ZIKV outbreak fits well with well described and recognized evidence of a period of transient heterotypic cross-protection (<1, 2 years) between DENV 1-4 serotypes[41–43]. Our results are consistent with short-lived, ZIKV-induced cross-protection which can suppress DENV circulation for 1-3 years depending on ZIKV attack rates and in agreement with previous modelling[21].

Incorporating immune interactions between distinct antigenic types is an ubiquitous practice in models of dengue transmission. Extensive theoretical work has shown that a diverse set of interactions, either in the form of cross-protection or infection enhancement, can replicate complex aspects of dengue population dynamics, including multiannual incidence patterns and serotype replacement[44,45]. Interestingly, it has been observed that similar dynamical patterns can also emerge without these interactions, solely due to stochastic effects and spatial segregation between hosts[46]. Despite the existence of these theoretical ambiguities, the adoption of different hypotheses regarding serotype interactions carries practical implications, particularly in the assessment of dengue vaccination impact[47,48]. In our model, we made the assumption that immune-mediated enhancement affects disease outcome exclusively and does not influence susceptibility to infection. Additionally, we assumed that all DENV-infected hosts contribute equally to transmission, irrespective of the number of preceding DENV infections. Available evidence suggests that post-secondary DENV infections are rarely reported[49,50], an aspect already accounted for in our model; however, their contribution to transmission remains unclear[51]. Our primary aim in this study was to investigate the interactions between ZIKV and DENV; however, future research work may expand upon the fundamental assumptions of our model to incorporate distinct immune interactions between DENV serotypes.

We calibrated our model to a hyperendemic scenario with continuous co-circulation of all four DENV serotypes. However, these viruses have been emerging and re-emerging in Brazil only in the last 40 years[52]. From re-emergence to hyperendemicity, the accumulation of immunity over multiple dengue epidemics has been proposed as an explanation for secular reductions in the age of reported cases[53]. Nonetheless, such a gradual process cannot fully explain the sudden changes observed between 2016 and 2019 in Brazil. It is well known that DENV circulation varies considerably across Brazil, with the Southeast and the Northeast reporting higher levels of transmission than other regions[18,54]. These differences translate in turn into different age patterns of illness across geographic locations[53]. Here, we partly accounted for heterogeneity in transmission through sensitivity analyses. We showed that regions with low levels of DENV transmission would be expected to display a smaller age-shift in DENV cases post-ZIKV compared to areas of high DENV transmission. On the one hand, this effect could help explain observed differences between states historically associated with low and high dengue transmission; on the other hand, the lack of seroprevalence data limits our ability to test this hypothesis. Similarly, a more complete historical picture of circulating dengue serotypes within each region or state would be helpful to investigate whether the outcome of ZIKV-DENV interactions is serotype-specific. However, serotyping is not performed within Brazilian states, and available data is insufficient. We note that our results, being calibrated in connection to a specific region of Brazil, support the occurrence of ZIKV-induced enhancement in DENV1 infections, since this was the serotype that dominated resurgence in 2018-2019 in the North and Northeast regions. This result is in line with data-supported observations that ZIKV previous infection enhances infection with DENV2[34], but contrasts previous predictions not yet supported by data of cross-protection against DENV1[55]. Finally, we stress that the model used and its outputs are at the population-level, from which it is not possible to identify or infer which specific immune mechanism is more likely to drive the proposed enhancement of DENV1 infections by ZIKV immunological responses.

In conclusion, we found that most Brazilian states experienced a reduction in the age of both DENV infections and clinical outcomes reported in 2018-2019 compared to the pre-ZIKV era. The observed age-shift across locations was statistically associated with the background ZIKV attack rate during 2015-2016 after its introduction in Brazil. We showed that a combination of immune interactions related

to history of individual exposure to the viruses could explain the findings related to the age-shift and the temporary disappearance of DENV following the ZIKV epidemic. These results support previous hypotheses of ZIKV modulating severity of DENV infections, and call for further investigations on the nature and consequences of cross-reactivity between these two flaviviruses. In addition, it would be interesting to investigate whether other areas in South and Central America that are hyperendemic to DENV and that experienced a large ZIKV outbreak in 2015-2016 experienced a similar variation in the DENV case age.

Immunological cross-reactivity in consecutive flavivirus infections is a well-known phenomenon. Flaviviruses include some of the most widespread and relevant mosquito-borne viruses for human and animal public health, including the dengue, Zika, West Nile, Usutu, St. Louis encephalitis, Japanese encephalitis, and Yellow Fever viruses. Knowledge on the background population-level seroprevalence of flaviviruses, as demonstrated in this study for ZIKV, is crucial for modelling and population-level research initiatives. However, seroprevalence data is virtually nonexistent, even for endemic viruses such as DENV in Brazil. We thus underscore the research and public health importance of investing in routinely monitoring seroprevalence of flaviviruses in endemic settings. Ultimately, a better understanding in this direction would contribute to the development of safe vaccines and timely characterization of individual and population-level disease risk upon future large epidemics of already circulating flaviviruses (e.g. of ZIKV once herd-immunity subsides) or of emerging ones.

## Methods

### Individual-based model

We implemented a stochastic, discrete-time individual-based model to simulate the spread of DENV ($D$) and ZIKV ($Z$) in a population of size $N$. Here, host life expectancy is modelled to mimic the underlying real age distribution of the population, using a Weibull-distributed with scale $\theta_a$ and shape $k_a$. Dying individuals are immediately replaced with newborns, which we assume to be immunologically naive to any circulating viruses.

Our model features 4 DENV serotypes, which we assume to share the same model parameters, and a single ZIKV serotype. A graphical illustration of the model can be found in Supplementary Fig. 9. Once infected, exposed hosts become infectious with rate $\epsilon_X (X = D, Z)$, and then recover with rate $\sigma_X$. Hosts infected with $X$ can transmit to any susceptible individual with rate $\beta_X(t)/N$, where $\beta_X(t)$ is the transmis-

sion rate at time $t$, which we assume to change periodically to account for seasonality:

$$\beta_X(t) = \beta_{0,X} \left[ 1 + \beta_{1,X} \cos \left( \frac{2\pi}{365} (t - T_X) \right) \right] \qquad (1)$$

where $\beta_{0,X}$ is the average transmissibility of $X$, $\beta_{1,X}$ is the strength of seasonal forcing, and $T_X$ indicates the timing of maximum transmissibility. For simplicity we do not model the vector component but take the incubation period to be the sum of the extrinsic and the intrinsic incubation periods to obtain temporal dynamics comparable with models including the vector.

We assume that each serotype (of DENV or ZIKV) grants lifelong complete protection against homotypic re-infection, while cross-protection against a different serotype is temporary, with mean duration $l_X$, and depends on an individual's immune history as well as the identity of the infecting virus (DENV1-4 or ZIKV). Temporary cross-protection induced by a DENV serotype prevents infection against challenge from a distinct DENV serotype, whereas cross-protection induced by DENV against ZIKV (and vice-versa) only modulates the transmission rate $\beta_Z (\beta_D)$ by a factor $\gamma_{DZ} (\gamma_{ZD})$. Finally, for simplicity we assume that co-infection is not possible.

### Model calibration

We simulate DENV and ZIKV dynamics in a community comprising $N = 300,000$ individuals in a context taylored to be similar to the state of Bahia. To this end, we initialise the host population so that it matches the age distribution and DENV age-seropositivity profile of its capital city Salvador in March 2015, i.e. before the emergence of ZIKV. Further details on model calibration and initialisation using data from a serological survey can be found in Supplementary Fig. 10 and Supplementary Methods. DENV and ZIKV parameters were informed by the literature where possible (see Table 1), while DENV transmission parameters were set to yield realistic incidence patterns in terms of seasonality and age structure (Supplementary Fig. 10 and 11).

### ZIKV introduction

Instead of simulating ZIKV explicitly, we set a varying proportion $\rho_Z$ of hosts as infected with the virus and set $\beta_{0,Z} = 0$, preventing onward transmission. In this way we can easily control the background attack rate of ZIKV ($\rho_Z$) as reported, while avoiding practical issues related to

## Table 1 | Model parameters with their description and values

| Name | Description | Value | References |
|------|-------------|-------|------------|
| $R_{0,D}$ | DENV basic reproductive number | 0.8–1.8 | Explored |
| $\epsilon_X$ | Incubation rate | 0.06 $d^{-1}$ (DENV & ZIKV) | 62 |
| $\sigma_X$ | Recovery rate | 0.22 $d^{-1}$ (DENV & ZIKV) | 62 |
| $\beta_{0,D}$ | Average DENV transmissibility | Computed as $R_{0,D} \cdot \sigma_D$ | – |
| $\beta_{1,D}$ | Amplitude of DENV seasonal forcing | 0.3 | Assumed |
| $T_X$ | Timing of maximum transmission | 15 $d$ (DENV) | Based on case data[58] |
| $l_D$ | Average duration of cross-protection induced by DENV infection | 2 y | 43 |
| $l_Z$ | Average duration of cross-protection induced by ZIKV infection | 0.25-1.25 y | Explored |
| $\gamma_{DZ}$ | Reduction in ZIKV transmissibility mediated by DENV (while cross-protected) | 1 | Assumed |
| $\gamma_{ZD}$ | Reduction in DENV transmissibility mediated by ZIKV (while cross-protected) | 0 | Assumed |
| $r_D$ | DENV introduction rate | 0.07 $d^{-1}$ | Assumed |
| N | Population size | 300000 | Assumed |
| $\theta_a$ | Scale Weibull lifetime distribution | 62 y | Based on population data[58] |
| $k_a$ | Shape Weibull lifetime distribution | 3.3 | Based on population data[58] |

The label $X = D, Z$ refers to DENV and ZIKV, respectively. Note that in the main analysis ZIKV is introduced artificially and is not allowed to transmit; hence $R_{0,Z} = \beta_Z = 0$.

direct simulation (Supplementary Fig. 12). Nonetheless, we show that both simulation schemes can yield similar DENV dynamics and uniform attack rates of ZIKV across age classes (Supplementary Fig. 12), in agreement with epidemiological observations[17] and with theoretical expectations for a novel emerging pathogen.

## Analysis of incidence age distributions

We used a bootstrap sampling strategy to estimate the mean case $\bar{A}$ and its uncertainty from age counts. Given a binned histogram with incident counts $\{n_i\}$ in age class $i = 1, \ldots, m$ with extrema $[a_i, b_i)$, we construct $n_{rep}$ bootstrapped histograms $\{n'_i\}$ with the same total counts $\sum_{i=1}^{m} n'_i = \sum_{i=1}^{m} n_i \equiv n$. Each histogram is obtained by sampling age classes $n$ times with replacement with sampling frequencies $\{n_i/n\}_{i=1,\ldots,m}$. Then, we draw $n'_i$ (integer) age values uniformly within the interval $[a_i, b_i)$ for each bin $i = 1, \ldots, m$. Using these samples, it is possible to compute any summary statistics, including mean case age $\bar{A}$ and the age-shift $\Delta = (\bar{A}_{post} - \bar{A}_{pre})/\bar{A}_{pre}$, where $\bar{A}_{pre}$ and $\bar{A}_{post}$ are calculated from the appropriate age distributions pre- and post-ZIKV, respectively. We used a two-sample permutation test to determine whether an age distribution $\{n_i^{(1)}\}$ was significantly younger than a second distribution $\{n_i^{(2)}\}$. Briefly, we merged counts $\{n_i^{(1)}\}$ and $\{n_i^{(2)}\}$ into a single distribution, which we used to resample $n_{rep} = 100000$ pairs of age distributions containing $n^{(1)}$ and $n^{(2)}$ points, respectively. We then calculated the difference between the corresponding mean ages using bin midpoints. The statistical significance of the observed age-shift was assessed based on the proportion of samples for which the mean age difference was smaller (more negative) than the observed value. Estimations in the main manuscript make use of all age bins except the last one, which corresponds to individuals aged 80 years and older.

## Regression analysis

We fitted a linear regression model to study the effect of a set of covariates on the age-shift $\Delta$ of DENV infections pre- and post-ZIKV. Covariates included ZIKV attack rate, the (square root of the) number of DENV cases per 100000 population, the proportion of DENV1 laboratory isolates during the 2018-2019 wave (versus other serotypes), and the average Index P before and after the emergence of ZIKV. We also considered a suite of linear models that either included or excluded ZIKV attack rate and with constant or region-specific intercepts. Model inference was realised through Bayesian Markov chain Monte Carlo using the *rstanarm* R package v2.21.3[56]. We compared these models in terms of their predictive accuracy using leave-one-out cross-validation, computed using the *loo* R package v2.5.1[57].

## Population data

We used official resident population estimates from 2001 to 2019 obtained from the Department of Informatics of the Unified Health System (DATASUS[58]).

## ZIKV incidence data

Weekly ZIKV notified cases (clinically suspected and confirmed) in Feira de Santana were obtained directly from the Secretaria Municipal de Saúde of the city. The clinical criteria included a pruritic maculopapular rash accompanied by at least one of the following symptoms: fever, conjunctival hyperemia or non-purulent conjunctivitis, arthralgia or polyarthralgia and periarticular edema. The confirmation criteria included those testing positive by RT-PCR or IgM serology, with tests performed at the Central Laboratory of Bahia (Laboratório Central da Bahia, LACEN-BA).

## Dengue incidence data

We collated dengue incidence counts obtained from the Brazilian Information System for Notifiable Diseases (SINAN) through the DATASUS[58]. We obtained yearly incidence of probable dengue cases from 2001 to 2019 by age group. We selected only cases that were confirmed by laboratory and/or clinical criteria. The breakdown of incidence by serotype was also obtained from the same source (from 2014 to 2019). Weekly DENV-notified cases (clinically suspected and confirmed) in the Bahian city of Feira de Santana were obtained directly from the Secretaria Municipal de Saúde of the city. The clinical criteria included fever for 2-7 days and two or more symptoms among: nausea, vomiting, rash, myalgia, arthralgia, headache, retro-orbital pain, petechiae, positive tourniquet test, leukopenia. The confirmation criteria included those testing positive by ELISA N1, RT-PCR or IgM serology, with tests performed at the Central Laboratory of Bahia (Laboratório Central da Bahia, LACEN-BA).

## Dengue hospitalisation data

We collated dengue hospitalisation counts obtained from the Unified Health System (SUS) through the DATASUS[58], and obtained yearly hospitalisations from 2001 to 2019 by age group. It should be noted that data from SUS and SINAN use different age groups. We considered all hospitalisations with ICD-10 codes A90 (dengue fever, classical dengue) and A91 (dengue hemorrhagic fever), i.e. those for which dengue was indicated as the direct cause.

## Estimated ZIKV attack rate

Estimates of ZIKV attack rate by state were obtained from[39].

## DENV seroprevalence data

Dengue serological data used to initialise the individual-based model was obtained from a serosurvey in Salvador, Bahia, in March 2015, i.e. right before the onset of the Zika epidemic[59].

## Transmission suitability index (Index P)

Index P is a suitability measure that quantifies the transmission potential of adult female *Aedes aegypti* mosquitoes in relation to a specific host-mosquito-pathogen system. Its mathematical formulation is based on a mechanistic model of ZIKV transmission[17], further developed and validated in several studies (e.g. in Nakase et al.[60]), and is given by:

$$P(U, T) = \frac{a^v(U)^2 \cdot \phi^{v \to h}(T) \cdot \phi^{h \to v}(T) \cdot \nu^v(T) \cdot \nu^h}{\mu^v(U,T) \cdot (\tau^h + \mu^h) \cdot (\nu^h + \mu^h) \cdot (\nu^v(T) + \mu^v(U,T))} \quad (2)$$

Where $U$ is relative air humidity, $T$ is temperature in Celsius, $a^v$ is mosquito biting rate, $\phi^{v \to h}$ and $\phi^{h \to v}$ are the probabilities of transmission per bite from mosquito to host and viceversa, $1/\nu^h$ and $1/\nu^v$ are host and mosquito (extrinsic) incubation periods, $1/\mu^h$ and $1/\mu^v$ are host and mosquito lifespans and $1/\tau^h$ is host (intrinsic) infectious period. We used satellite climate data from Copernicus.eu and the *MVSE* package V1.01 to estimate Index P[61].

## Reporting summary

Further information on research design is available in the Nature Portfolio Reporting Summary linked to this article.

# Data availability

Counts of probable dengue virus cases and hospitalizations for individual states and the city of Salvador, Bahia, were obtained from the Brazilian Information System for Notifiable Diseases (SINAN, https://datasus.saude.gov.br/acesso-a-informacao/doencas-e-agravos-de-notificacao-2001-a-2006-sinan and https://datasus.saude.gov.br/acesso-a-informacao/doencas-e-agravos-de-notificacao-de-2007-em-diante-sinan) and Unified Health System (SUS) through the DATASUS (https://datasus.saude.gov.br/acesso-a-informacao/morbidade-hospitalar-do-sus-sih-sus). Population size estimates were obtained from DATASUS (https://datasus.saude.gov.br/populacao-residente). Weekly DENV and ZIKV notified cases (clinically suspected and

confirmed) in the Bahian city of Feira de Santana were obtained directly from the Secretaria Municipal de Saúde of the city (https://www.feiradesantana.ba.gov.br/servicos.asp?id=14&link=sms/vigilancia_saude/vigilancia_epidemiologica.asp). All relevant data used in Figs. 1–3 and Supplementary Fig. 1-8,14,16 are available in the GitHub repository https://github.com/francescopinotti92/dengue_and_zika_brazil.

## Code availability

Code necessary to reproduce the analyses in this study is available in the GitHub repository https://github.com/francescopinotti92/dengue_and_zika_brazil.

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

## Acknowledgements

F.P. is funded by the UKRI GCRF One Health Poultry Hub (Grant No. BB/S011269/1), one of twelve interdisciplinary research hubs funded under the UK government's Grand Challenge Research Fund Interdisciplinary Research Hub initiative. M.G. is funded by PON 'Ricerca e Innovazione' 2014-2020. M.R. is supported by a Human Frontiers Science Programme award (RGP018/2023). The authors would like to acknowledge the use of the University of Oxford Advanced Research Computing (ARC) facility in carrying out this work.

## Author contributions

F.P. and J.L. conceived and designed the study. F.P. and M.M.L. collated the data. F.P. analysed the data and performed simulations. F.P. and J.L. prepared the visualisations. All authors (F.P., M.G., M.M.L., E.M.C., L.C.J.A., S.G., M.R. and J.L.) interpreted the results. F.P., J.L., M.R. and M.G. wrote the original draft. F.P., J.L., M.R. and M.G. wrote the revised manuscript.

## Competing interests

The authors declare no competing interests.
