## [Peer Review File · Nature Communications]

Shifting patterns of dengue three years after Zika virus emergence in BrazilREVIEWER COMMENTS

Reviewer #1 (Remarks to the Author):

In the proposed manuscript the authors explore the changing patterns of dengue and Zika incidence in Brazil, in particular as a function of age group. They hypothesise that changing patterns of age-specific incidence could help us understand the immune interactions between the two viruses. DENV and ZIKV immunity is complex, and mathematical models are well placed to better evaluate their interaction.

Overall, I found the paper well written with a sound simulation framework, and with interesting results. My only major comment is that the entire premise of the paper is based on the observation in a shift of ages in DENV cases following the ZIKV epidemic. However, even in the absence of ZIKV, the age distribution of DENV cases in Brazil has not been stable (<https://doi.org/10.1371/journal.pntd.0000935>). The transition to endemicity since the reintroduction of the virus is shifting the age distribution younger, while reducing birth rates push the age of cases older. Depending on the local DENV epidemiology and demographic transitions, there will be different shifts in ages per year across the country. It is impossible to consider the impact of ZIKV on the age distribution of DENV cases, without also considering the broader age trends among DENV cases. While I appreciate that the authors include a discussion of this issue, it would be far more compelling if the authors at the very least present the trends in the ages of DENV cases prior to the ZIKV epidemic, and to statistically check that ZIKV really did significantly shift the age of cases.

Reviewer #2 (Remarks to the Author):

Overview

This was a really nice paper. This is a challenging question and topic to investigate given the complex (mostly unobserved) dynamics in Brazil and other areas hyperendemic to multiple co-circulating arboviruses. At the same time, it's a very important question and will likely continue to evade any kind of final conclusion until the field can obtain some kind of long-term, longitudinal sampling data from a cohort study. I do think that there needs to be further exploration and explanation of the modulation on DENV primary infections in the ZIKV-positive population. Thanks!

Major comments

- The authors suggest that the most plausible scenario from the model (supported by the data) is that in ZIKV-positive individuals (but naïve to DENV) some sort of immunomodulation is occurring to primary and possibly secondary DENV infections, therefore increasing DENV incidence in primary infections that are usually asymptomatic. I think this makes sense, but I think it needs more evidence from models or data. For example, the data in Figure 2 for Salvador is quite compelling. But for the large increase in hospitalizations for 1-4 and 5-9 year-olds, the conclusions implies that those individuals should have had ZIKV prior to 2018-2019 (i.e., when they were infants / toddlers). This could be further supported by annual force of infections estimates applied to different birth cohorts, i.e, the probability that someone born in 2014 was infected with ZIKV in 2015 then DENV in 2018/2019.
- An additional comment somewhat piggy-backing on the previous comment. If the conclusion that modulation is occurring on primary DENV infections in ZIKV positive individuals is true, would we see increase in older age group for DENV incidence in other areas of Brazil that are not hyperendemic for DENV, but still experienced a ZIKV epidemic in 2014-2015? I think this warrants more exploration.
- Another point regarding this same idea. Is there some kind of exploration or analysis that you could do to test the hypothesis that this is affecting cases vs. infection? For example, if the real thing going on here is ZIKV-modulation leading to an increase in symptomatic primary DENV infections, we could assume that DENV infections would be the same but that reported dengue cases would differ. Or are you suggesting modulation is happening at the immune level such that ZIKV is causing higher susceptibility among individuals such that DENV infections are actually

increasing but reporting remains the same? Please flesh this out further.

Minor comments

- I strongly suggest making the proportion of DENV/ZIKV incidence over age groups a bar plot instead of a line plot (e.g., Figure 2, Figure S5), possibly a paired barplot?. With the lines, it makes it difficult to tell which age group has the difference proportional values.
- Are there any concerns about misdiagnosis (btwn the arboviruses) w/in the data that you used?
- I know that the Nat Comm abstract template is short, but I think the abstract is too dense. It's hard to get the main message from it. Consider reworking it to some degree.
- Given the interesting conclusion regarding age-stratified differences in dengue incidence, I think the introduction needs more overview of regional force of infections and the age-stratified probability of 1, 2, 3+ DENV infections in hyperendemic areas. This could then be used as evidence for the dynamics in Santander.
- Did you consider the variations in the assumed attack rate? Did you use the median estimate from Moore et al? It is not necessary to explore this further, I am just curious how sensitive the results would be to variation in the attack rate.
- Is there some kind of formula for the Transmission Suitability Index? Can you include this?
- Figure 3 is very convincing and clear. Nice job on this visualization.
- Are there data from other areas hyperendemic to dengue that also had a Zika epidemic (e.g., Thailand, Cambodia, Colombia?) that show a similar trend? I'm wondering if this is site-specific or a more general finding.

Dear Editor and Referees,

We thank the referees for taking the time to review our manuscript and for their positive assessment. We believe that both referees raised important points that improved the quality of our analyses. We provide below a detailed discussion of each point, with our answers highlighted in blue. We attempted to make our response self-contained by reporting main changes and additions to both text and figures here. In addition, we provide copies of our revised manuscript and supplementary material with all main changes being colour-highlighted. Specifically, text additions/modifications are highlighted in blue, while removed text is highlighted in red and with a strikethrough.

To ensure clarity and facilitate a thorough review, we attempted to make this response self-contained. We reported the main modifications and additions to both the manuscript text and figures within this document. Furthermore, we have attached copies of the revised manuscript and supplementary materials, where significant changes have been thoughtfully highlighted using blue for additions/modifications and red for removed text.

REVIEWER COMMENTS

Reviewer #1 (Remarks to the Author):

In the proposed manuscript the authors explore the changing patterns of dengue and Zika incidence in Brazil, in particular as a function of age group. They hypothesise that changing patterns of age-specific incidence could help us understand the immune interactions between the two viruses. DENV and ZIKV immunity is complex, and mathematical models are well placed to better evaluate their interaction.

Overall, I found the paper well written with a sound simulation framework, and with interesting results. My only major comment is that the entire premise of the paper is based on the observation in a shift of ages in DENV cases following the ZIKV epidemic. However, even in the absence of ZIKV, the age distribution of DENV cases in Brazil has not been stable (<https://doi.org/10.1371/journal.pntd.0000935>). The transition to endemicity since the reintroduction of the virus is shifting the age distribution younger, while reducing birth rates push the age of cases older. Depending on the local DENV epidemiology and demographic transitions, there will be different shifts in ages per year across the country. It is impossible to consider the impact of ZIKV on the age distribution of DENV cases, without also considering the broader age trends among DENV cases. While I appreciate that the authors include a discussion of this issue, it would be far more compelling if the authors at the very least present the trends in the ages of DENV cases prior to the ZIKV epidemic, and to statistically check that ZIKV really did significantly shift the age of cases.

We are pleased that the reviewer valued our work and we are thankful for their comments.

We agree that it is useful to frame our findings in the context of broader age trends of DENV cases. Indeed, as correctly pointed out by the reviewer, secular trends in DENV circulation and host demographics act together to change the age distribution of DENV cases over time (as described in Rodriguez-Barraquer et al, e.g.). It is possible that such age trends may result in age-shifts consistent with observations without the need to invoke ZIKV-mediated interactions. We therefore performed further statistical analyses in order to disentangle the role of DENV age trends.

We fitted a Gaussian Process regression model (GP) to yearly mean case age \bar{A}_t from $t = 2001$ to 2015 , i.e. before ZIKV emerged, using time and spatial coordinates as covariates; this allowed us to obtain a non-parametric estimate of trends in mean case age across time and space, and make projections of \bar{A}_t for $t = 2018$ and 2019 (see the new Supplementary Fig. 8 and 9, attached below as well). We find that the GP model fails to explain the decline in mean age DENV in states that coincidentally experienced larger ZIKV attack rates. In fact, the model predicts a positive correlation between age-shift and ZIKV attack rate (see new Supplementary Fig. 10, which is attached below), at variance with observations reported in Fig. 3 in the main manuscript. Noting that the GP model is oblivious to ZIKV attack rates, these findings indicate that age trends of DENV prior to ZIKV alone are unlikely to solely explain observations. They instead suggest a potential role of some other driver, such as the emergence of ZIKV, as already confirmed by previous statistical analyses. These additional analyses are now included in three new supplementary figures, which we reference in the main text:

"Additionally, we find that the negative correlation between estimated ZIKV attack rate and Δ is robust with respect to the pre-ZIKV reference period used to compute the mean age of DENV \bar{A}_{pre} (Supplementary Fig. 7), and that it can not be explained by secular changes in mean age of DENV (This analysis can be found in Supplementary Fig. 8,9,10)."

We also calculated yearly birth rates by state, displayed in Additional Fig. 1 below. Birth rates have been declining almost everywhere in the last 20 years, the overall decline over the period under investigation is too small to account for the observed age-shift. Also, it is difficult to identify a spatial trend that could help explain observed variation in age-shift across states. For this reason, we prefer to leave Additional Fig. 1 in this document.

Supplementary Figure 8. **Time series analysis of mean age of DENV cases.** We use gaussian process regression to investigate spatio-temporal trends in the mean age of DENV cases, $\bar{A}_{i,t}$, from $t = 2001$ to $t = 2019$ and across states $i = 1, \dots, 27$. Gaussian processes represent a flexible, non-parametric framework to uncover non-linear trends in data. We assume that the logarithm of mean case age $\hat{A}_{i,t} = \log \bar{A}_{i,t}$ is distributed according to a gaussian process with zero mean ($\hat{A}_{i,t}$ is also standardised by subtracting its mean prior to fitting) and covariance function $C \cdot K[\vec{u}_k, \vec{u}_l] + \sigma^2 \delta_{k,l}$, where C is a constant, \vec{u}_k is a vector of covariates consisting of space and time coordinates for observation k and $\sigma^2 \delta_{k,l}$ is a white noise covariance function. Spatial coordinates were calculated for each state using the *representative_point* function in *Geopandas* v0.11.1. We choose K to be a radial basis function kernel with a separate scale parameter for each covariate (3 in total). In order to verify whether pre-ZIKV trends in mean case age of DENV can explain post-ZIKV data, we fit this model to data up to $t = 2015$ (grey scatters) and use the resulting posterior predictive distribution to predict values $\bar{A}_{i,t}$ for $t > 2015$ (red scatters). Lines and shaded areas denote the posterior median and 95% C.I. based on 400 samples from the posterior predictive distribution. Further comments on these results can be found in Supplementary Fig. 10.

Supplementary Figure 9. **Time series analysis of mean age of DENV hospitalisations.** This figure repeats the analysis of Supplementary Fig. 8 but using hospitalisation data instead of cases. Further comments on these results can be found in Supplementary Fig. 10.

Supplementary Figure 10. **Pre-ZIKV trends of mean age of DENV do not explain the age-shift in 2018-2019.** In this figure we analyse short-term projections from the gaussian process model fitted in Supplementary Fig. 8 and 9 (left and right columns, respectively). More in detail, we use the gaussian process posterior predictive distribution to sample $\bar{A}_{i,2018}$ and $\bar{A}_{i,2019}$ and calculate the age-shift $\Delta_{gp,i}$. This process is repeated 5000 times, and model residuals are calculated by subtracting observed values $\Delta_{obs,i}$. For simplicity, we calculate \bar{A}_{pre} and \bar{A}_{post} by simply taking the mean value of mean age of DENV over the periods 2013-2015 and 2017-2018, respectively. Median values (scatters) and 95% C.I. (bars) of model residuals for each state are shown in panels A, B. The colour code is the same as in Fig. 3 in the main manuscript. We find that the model overestimates the age-shift in states that experienced large ZIKV outbreaks, and that this discrepancy increases in magnitude with ZIKV attack rate. Panels C, D further elaborate on this trend by showing the distribution of Spearman's rank correlation coefficient between Δ_{gp} and ZIKV attack rates. The model actually predicts a strong positive correlation between these variables, at variance with observations. These results suggest that pre-ZIKV trends of mean age of DENV can not explain age patterns observed during DENV resurgence in 2018 and 2019.

Additional Figure 1. **Yearly birth rates in Brazil.** Each panel groups state-level yearly birth rates per 1000 population for a single region in Brazil. Data on new births were downloaded from DATASUS, i.e. from the same source of population size data.

Reviewer #2 (Remarks to the Author):

Overview

This was a really nice paper. This is a challenging question and topic to investigate given the complex (mostly unobserved) dynamics in Brazil and other areas hyperendemic to multiple co-circulating arboviruses. At the same time, it's a very important question and will likely continue to evade any kind of final conclusion until the field can obtain some kind of long-term, longitudinal sampling data from a cohort study. I do think that there needs to be further exploration and explanation of the modulation on DENV primary infections in the ZIKV-positive population. Thanks!

We thank the reviewer for their positive assessment of our manuscript and their comments/suggestions.

Major comments

- The authors suggest that the most plausible scenario from the model (supported by the data) is that in ZIKV-positive individuals (but naïve to DENV) some sort of immunomodulation is occurring to primary and possibly secondary DENV infections, therefore increasing DENV incidence in primary infections that are usually asymptomatic. I think this makes sense, but I think it needs more evidence from models or data. For example, the data in Figure 2 for Salvador is quite compelling. But for the large increase in hospitalizations for 1-4 and 5-9 year-olds, the conclusions implies that those individuals should have had ZIKV prior to 2018-2019 (i.e., when they were infants / toddlers). This could be further supported by annual force of infections estimates

applied to different birth cohorts, i.e, the probability that someone born in 2014 was infected with ZIKV in 2015 then DENV in 2018/2019.

We thank the reviewer for this important comment. To flesh this out further, we retrieved estimates of yearly forces of infection for the North-East region from Brito et al., *Nature Communications*, 2021, and adapted their methodology to calculate the probability of being infected with DENV in 2018 or 2019 by birth cohort (up to 9 year-olds) and by ZIKV exposure status. More precisely, we calculated the probability of primary and secondary DENV infections in both ZIKV-negative and ZIKV-positive hosts. We assumed that the probability P_z of being infected with ZIKV was independent of age (as done also in the rest of the manuscript) and considered two cases with $P_z = 0.36$ and $P_z = 0.77$; these values correspond to the smallest and the largest ZIKV attack rates in Northeastern states. For simplicity we assumed that hosts could get infected with ZIKV during 2016 only, i.e. during the bulk of the epidemic. The results are shown in the new Supplementary Fig. 16 (attached below). In both 2018 and 2019, the probability of DENV infection in ZIKV-positive hosts is non-negligible when compared to ZIKV-negative hosts already for $P_z = 0.36$. This demonstrates that, according to available evidence, it would make sense for ZIKV-induced immunomodulation to have an effect on children less than 10 years old. Moreover, the figure also shows that primary DENV infections are more likely than secondary infections in this age group, suggesting that if some degree of immunomodulation occurs in ZIKV-positive hosts, it should affect primary DENV infections to some extent. These results are now mentioned in the Results section and are put into the context of the relative increase in paediatric cases shown in Fig. 2:

"Immunomodulation in primary DENV infections would also explain the large contribution of paediatric infections (children aged 1-9 years old) to hospitalisations as showcased in Fig. 2. Indeed, a further analysis suggests that a significant fraction of incident cases reported in 2018 and 2019 in this age group in the Northeast are expected to be primary DENV infections that had been exposed to ZIKV in 2015-2016 (Supplementary Fig. 16)."

Supplementary Figure 16. **Reconstructing seroincidence in children in the Northeast.** Yearly estimates of the force of infection λ_t of DENV from $t = 2002$ to 2019 in the Northeast of Brazil. We recovered λ_t from the second panel in Fig. 3 in Brito et al. using *PlotDigitizer*. (B-I) Probability that children aged between $a = 0$ and 9 years old experience a primary or a secondary DENV infection in 2018 (B-E) or 2019 (F-I), broken down by whether they had been previously infected with ZIKV or not. Each panel shows infection probabilities for two scenarios with a ZIKV attack rate of 0.36 (fuchsia) or 0.77 (gray), i.e. the extremal values reported for individual Northeastern states in Fig. 3 in the main manuscript. Panels (B-I) show that primary DENV infections are more common than secondary

infections in ZIKV-positive children aged less than 10 years old in areas with high ZIKV transmission. We calculated infection probabilities by adapting the methodology presented in Brito et al.. Let $P_{a,t}(D|Z = 0, 1)$ denote the probability of primary ($D = 1$) or secondary ($D = 2$) DENV infections in hosts aged a years old in year t , conditional on $Z = 0, 1$ past ZIKV infections. This quantity is calculated as $P_{a,t}(D = 1|Z) = (1 - e^{-4\lambda_t}) \cdot d_{a,t}^{(0)} \cdot z_{a,t}^Z \cdot (1 - z_{a,t})^{1-Z}$, where $d_{a,t}^{(0)}$ is the probability of having experienced no DENV infections at age a and $z_{a,t}$ is the probability of having experienced a ZIKV infection at age a in year t . We assume that $z_{a,t}$ equals ZIKV attack rate, if the host was born in 2016 or earlier, and is 0 otherwise. This assumption implies uniform attack rates over age (as assumed also in the rest of the manuscript). $P_{a,t}(D = 2|Z)$ is defined similarly, but with the infection term being $1 - e^{-3\lambda_t}$ and $d_{a,t}^{(0)}$ replaced by $d_{a,t}^{(1)}$, namely the probability of having experienced exactly one DENV infection at age a . We calculate $d_{a,t}^{(0)}$ and $d_{a,t}^{(1)}$ as $d_{a,t}^{(0)} = \prod_{i=t-a}^{t-1} e^{-4\lambda_i}$ and $d_{a,t}^{(1)} = \sum_{j=t-a}^{t-1} \left(\prod_{i=t-a}^{j-1} e^{-4\lambda_i} \right) (1 - e^{-3\lambda_j}) \left(\prod_{i=j+1}^{t-1} e^{-3\lambda_i} \right)$ for $a > 0$ and $d_{a,t}^{(0)} = 1, d_{a,t}^{(1)} = 0$ for $a = 0$.

- An additional comment somewhat piggy-backing on the previous comment. If the conclusion that modulation is occurring on primary DENV infections in ZIKV positive individuals is true, would we see increase in older age group for DENV incidence in other areas of Brazil that are not hyperendemic for DENV, but still experienced a ZIKV epidemic in 2014-2015? I think this warrants more exploration.

We thank the reviewer for this comment. We believe that this is a reasonable conclusion given our hypotheses. Indeed, Supplementary Fig. 14 in the initial submission partially confirms this conclusion by showing how the mean age of observed cases varies as a function of DENV's basic reproductive number $R_{0,D}$, with and without a ZIKV epidemic. Clearly, lower values of $R_{0,D}$ correspond to higher mean age of infection, which is compatible with areas historically associated with low DENV transmission displaying older age distributions of reported cases. In order to better elucidate the potential impact of ZIKV-induced modulation of disease on the age distribution of DENV in areas with varying levels of DENV transmission, we expanded Supplementary Fig. 14 (now Supplementary Fig. 18, attached below) with age distributions of reported DENV cases for $R_{0,D} = 1.2, 1.4, 1.6, 1.8$, corresponding to a mean age of DENV decreasing from 31 to 23 years (panel A, dashed line). We find that the bulk of the age distribution always shifts to younger ages, but the modal age group is always the one from 5 to 15 years. This effect is the consequence of ZIKV-mediated enhancement of disease in primary DENV infections, which are increasingly abundant in younger children.

These results are now mentioned in the Results section:

"Interestingly, the shifted age distributions always peak in the 5-14 years old under mechanism (1) for a range of $R_{0,D}$ values (Supplementary Fig. 18). However, older age groups also become more pronounced in low-transmission settings, where the baseline mean age of DENV cases is higher."

Supplementary Figure 18. **Impact of baseline DENV transmission.** (A) Mean observed age of incident cases as a function of $R_{0,D}$ and for different values of ZIKV attack rate (\bar{A}_{post}). The dashed line corresponds to simulations without ZIKV and is thus equivalent to \bar{A}_{pre} , the mean observed case age before ZIKV is introduced. The mean observed case age before (\bar{A}_{pre}) and after (\bar{A}_{post}) ZIKV, which enter Δ 's definition through $\Delta = \bar{A}_{post}/\bar{A}_{pre} - 1$, do not depend on $R_{0,D}$ in the same way. Typically, \bar{A}_{post} decreases with $R_{0,D}$ (solid lines), while \bar{A}_{pre} initially increases with $R_{0,D}$ and then declines (dashed line). Taken together, these results mean that Δ must decrease more rapidly at low $R_{0,D}$. (B-E) Age distributions of reported DENV cases for different values of $R_{0,D}$, without ZIKV (grey) and with a ZIKV epidemic infecting a proportion $\rho_Z = 0.6$ of the host population (fuchsia). Panels (B-E) show that the effect of ZIKV-induced modulation of disease would be to shift the entire distribution to younger ages; moreover, the age group from 5 to 15 years is always the modal one. Results are based on the assumption that ZIKV enhances disease in primary DENV infections only (mechanism (1) in the main text). To obtain these plots, we let DENV spread for a burn-in period of 100 years before ZIKV is introduced at $t = 0.45$ y. We compute DENV incidence during the fourth year since ZIKV emergence and average results over 100 independent simulations. Other parameters are set to default values.

• Another point regarding this same idea. Is there some kind of exploration or analysis that you could do to test the hypothesis that this is affecting cases vs. infection? For example, if the real thing going on here is ZIKV-modulation leading to an increase in symptomatic primary DENV infections, we could assume that DENV infections would be the same but that reported dengue cases would differ. Or are you suggesting modulation is happening at the immune level such that ZIKV is causing higher susceptibility among individuals such that DENV infections are actually increasing but reporting remains the same? Please flesh this out further.

We thank the reviewer for this comment. In this work, we assume that ZIKV infection affects DENV at two levels:

- A. In the short term, ZIKV-induced cross-reactivity does not affect susceptibility against DENV infection but prevents onward transmission. To put it simply, we assumed that hosts with a recent ZIKV infection could still become infected with DENV (with the same susceptibility) but could not transmit due to cross-immunity.
- B. In the medium/long term, ZIKV-induced cross-reactivity affects severity of disease only (and not susceptibility to DENV) and hence the odds of a case

becoming reported (e.g. because ZIKV increases the odds of a symptomatic infection).

We find that these mechanisms together are able to explain certain patterns of DENV epidemiology in Brazil, namely the decline of DENV in 2016-2017 and the age of reported DENV cases in 2018-2019. As the reviewer correctly suggests, it is possible that other manifestations of ZIKV-induced cross-reactivity, such as modified susceptibility to infection (without affecting severity), may as well explain these patterns. Comparing multiple hypotheses about immune-mediated interactions between ZIKV and DENV would require understanding the distinctive dynamical "traces" left by each mechanism on epidemiological data (e.g. incidence). In practice, however, untangling distinct mechanisms is a challenging task (Shrestha et Al, PLoS Computational Biology, 2011; Mair et Al, PLoS Computational Biology, 2019; Waterlow et Al, Epidemics, 2021; Man et Al, Journal of the Royal Society Interface, 2023) that merits a separate investigation. We now acknowledge this issue in the Discussion section:

"Future research work is required, however, to verify whether alternative mechanisms, e.g. enhancement of infection rather than disease, provide a better explanation of empirical observations."

Minor comments

- I strongly suggest making the proportion of DENV/ZIKV incidence over age groups a bar plot instead of a line plot (e.g., Figure 2, Figure S5), possibly a paired barplot?. With the lines, it makes it difficult to tell which age group has the difference proportional values.

We modified these figures accordingly to display paired bar plots and make differences between age groups more apparent. In addition, we added the variation in mean age between 2014-2015 and 2018-2019 as insets in Fig. 2, in the hope to make the trend (i.e. the overall shift to younger ages) clearer to the reader.

- Are there any concerns about misdiagnosis (btwn the arboviruses) w/in the data that you used?

The reviewer is right in pointing out that some misdiagnosis between DENV, CHIKV and ZIKV might have happened due to similar clinical manifestations and imperfect tests. Previous investigations (R J Oidtman et al., PLoS Neglected Tropical Diseases, 2021) suggest that most misdiagnoses are likely ZIKV cases that were classified as DENV. Furthermore, most misdiagnoses should have happened between late 2015 and early 2016, at the height of the ZIKV epidemic. Given these findings, we believe that misdiagnoses should be limited in the case of DENV incidence data as we focussed on the periods 2013-2015 and 2018-2019. On the other hand it is possible that ZIKV attack rate estimates from Moore et al. are lower than the real attack rates. However, it is not clear if ZIKV underestimation is spatially heterogeneous, which could affect correlation

with DENV age-shifts. Moore et al. estimates of ZIKV attack rates are the most comprehensive and up to date (as best as we know), offering a unique opportunity to test a wide range of epidemiological hypotheses. We have now mentioned the issue of misdiagnosis between arboviruses in the following sentences in the Discussion section (citing also Oidtman et al.'s article):

"It is possible, however, that current estimates of ZIKV attack rates underestimate true ZIKV circulation due to misdiagnosis between Zika and dengue (Oidtman et al, 2021). Despite these concerns, our results should still hold provided that the extent of underestimation is not extremely heterogeneous across Brazil."

- I know that the Nat Comm abstract template is short, but I think the abstract is too dense. It's hard to get the main message from it. Consider reworking it to some degree.

We thank the reviewer for pointing this out. Accordingly, we reworked the abstract to make it less dense and better convey our main messages. The abstract now reads:

"In 2015, the Zika virus (ZIKV) emerged in Brazil, leading to widespread outbreaks in Latin America. Following this, many countries in these regions reported a significant drop in the circulation of Dengue virus (DENV), which resurged in 2018-2019. We examine age-specific incidence data to investigate changes in DENV epidemiology before and after the emergence of ZIKV. We observe that incidence of DENV was concentrated in younger individuals during resurgence compared to 2013-2015. This trend was more pronounced in Brazilian states that had experienced larger ZIKV outbreaks. Using a mathematical model, we show that ZIKV-induced cross-protection alone, often invoked to explain DENV decline across Latin America, cannot explain the observed age-shift without also assuming some form of disease enhancement. Our results suggest that a sudden accumulation of population-level immunity to ZIKV could suppress DENV and reduce the mean age of DENV incidence via both protective and disease-enhancing interactions."

- Given the interesting conclusion regarding age-stratified differences in dengue incidence, I think the introduction needs more overview of regional force of infections and the age-stratified probability of 1, 2, 3+ DENV infections in hyperendemic areas. This could then be used as evidence for the dynamics in Santander.

We thank the reviewer for raising this point. Indeed, heterogeneity in DENV transmission across Brazil is a major point that we strived to address in the main manuscript, as outlined also in the reply to the second major comment. However, given the general aim of the study, we would prefer to keep the focus of the Introduction section on ZIKV, and on potential immunological mechanisms that may lead to interactions with DENV. Nonetheless, we expanded one paragraph in the Discussion section to include a brief statement on regional differences in transmission across Brazil, and its consequences on age patterns of reported cases:

"It is well known that DENV circulation varies considerably across Brazil, with the Southeast and the Northeast reporting higher levels of transmission than other regions [Brito et al, Nature Communications, 2021; Bosco Siqueira Junior et al, International Journal of Infectious Diseases, 2022]. These differences translate in turn into different age patterns of illness across geographic locations [Rodriguez-Barraquer et al, PLoS Neglected Tropical Diseases, 2011]."

- Did you consider the variations in the assumed attack rate? Did you use the median estimate from Moore et al? It is not necessary to explore this further, I am just curious how sensitive the results would be to variation in the attack rate.

We confirm that we used only the median estimate from Moore et al, but the reviewer is right in pointing out that this estimate comes with its own statistical uncertainty. In order to assess the impact of such uncertainty, we used Moore et al's code and data (https://github.com/mooresea/Zika_IAR/tree/master) to explore the joint posterior distribution of state-level ZIKV attack rates. We generated 10000 samples from the posterior distribution, each sample including a value of ZIKV attack rate for each state. We then calculated Spearman correlation between ZIKV attack rates and observed age-shifts for each posterior sample. The distribution of correlation coefficients summarises the impact of estimation uncertainty on observed correlations and is shown in Additional Fig. 2. This figure provides further support for the negative relationship between ZIKV attack rate and age-shift reported in the main manuscript.

Additional figure 2. **Impact of estimation uncertainty on ZIKV attack rates.** Each panel shows the distribution of Spearman rank correlation coefficient between state-level dengue age-shifts (left: cases; right: hospitalisations) and ZIKV attack rates. A single correlation coefficient is calculated for each posterior sample for Moore et al's model.

- Is there some kind of formula for the Transmission Suitability Index? Can you include this?

We agree with the reviewer that the definition of the Transmission Suitability Index (Index P) should be included in the manuscript for completeness. We have now included

a short description and the mathematical expression defining the Index P in the main text.

- Figure 3 is very convincing and clear. Nice job on this visualization.

We thank the reviewer for their very kind comment.

- Are there data from other areas hyperendemic to dengue that also had a Zika epidemic (e.g., Thailand, Cambodia, Colombia?) that show a similar trend? I'm wondering if this is site-specific or a more general finding.

This is an important point and indeed it would be interesting to test our hypothesis in the context of other dengue hyperendemic areas in South and Central America, where ZIKV rampaged in 2015-2016. Our initial question originated from an analysis of age-stratified data for the city of Feira de Santana, which we later expanded to other areas in Brazil. We searched for age-stratified dengue case data across governmental websites and in published manuscripts but we could not easily access data analogous to Brazil. Authors from <https://www.ncbi.nlm.nih.gov/pmc/articles/PMC6579519/> provide links to several official sources of data but none of these links work anymore. Hopefully, data for Ecuador should be released soon by the authors of the following preprint <https://www.medrxiv.org/content/10.1101/2023.05.25.23290519v1.full.pdf>. To emphasise the importance of such investigations, we included the following sentence in the Discussion section:

"In addition, it would be interesting to investigate whether other areas in South and Central America that are hyperendemic to DENV and that experienced a large ZIKV outbreak in 2015-2016 experienced a similar variation in the DENV case age."

REVIEWERS' COMMENTS

Reviewer #1 (Remarks to the Author):

The authors have been very responsive to my concerns. I believe the additional analyses - especially the long term trends in ages of cases really strengthen the paper.

Reviewer #2 (Remarks to the Author):

The authors did a fantastic and very thorough job in responding to my comments and suggestions. I have no further comments except to say that I think this is a fantastic manuscript and will spur further research. Thank you!